# Insights into a Cancer-Target Demethylase: Substrate Prediction through Systematic Specificity Analysis for KDM3A

**DOI:** 10.3390/biom12050641

**Published:** 2022-04-27

**Authors:** Anand Chopra, William G. Willmore, Kyle K. Biggar

**Affiliations:** 1Institute of Biochemistry, Carleton University, Ottawa, ON K1S 5B6, Canada; anandchopra@cmail.carleton.ca; 2Department of Biology, Carleton University, Ottawa, ON K1S 5B6, Canada

**Keywords:** lysine demethylase, KDM3A, substrate specificity, motif, non-histone, substrate prediction

## Abstract

Jumonji C (JmjC) lysine demethylases (KDMs) catalyze the removal of methyl (-CH_3_) groups from modified lysyl residues. Several JmjC KDMs promote cancerous properties and these findings have primarily been in relation to histone demethylation. However, the biological roles of these enzymes are increasingly being shown to also be attributed to non-histone demethylation. Notably, KDM3A has become relevant to tumour progression due to recent findings of this enzyme’s role in promoting cancerous phenotypes, such as enhanced glucose consumption and upregulated mechanisms of chemoresistance. To aid in uncovering the mechanism(s) by which KDM3A imparts its oncogenic function(s), this study aimed to unravel KDM3A substrate specificity to predict high-confidence substrates. Firstly, substrate specificity was assessed by monitoring activity towards a peptide permutation library of histone H3 di-methylated at lysine-9 (i.e., H3K9me2). From this, the KDM3A recognition motif was established and used to define a set of high-confidence predictions of demethylation sites from within the KDM3A interactome. Notably, this led to the identification of three in vitro substrates (MLL1, p300, and KDM6B), which are relevant to the field of cancer progression. This preliminary data may be exploited in further tissue culture experiments to decipher the avenues by which KDM3A imparts cancerous phenotypes.

## 1. Introduction

Lysine methyltransferases (KMTs) and lysine demethylases (KDMs) dynamically regulate the lysine methylation status of a broad range of proteins [1,2]. In this manner, these modifying enzymes are fundamentally tied to many physiological and pathophysiological processes, such as cancer progression and therapeutic resistance [3,4,5]. Specifically, the biological roles of these methyl-regulators have been primarily tied to their ability to alter gene expression through the modulation of histone methylation status. As research progressed and techniques studying lysine methylation continued to expand, it became increasingly evident that lysine methylation occurring beyond histone proteins is intimately tied to the biological roles of these modifying enzymes.

Although research on the role of non-histone methylation in cancer is still in its infancy, most of the research has primarily been focused on KMTs. In fact, the first KMT inhibitor to be given FDA approval occurred recently; tazemetostat is an inhibitor of the EZH2 methyltransferase and was approved for the treatment of epithelioid sarcoma [6]. Several KDMs are now known to be promising cancer drug targets due to their biological roles in cancer progression and therapeutic resistance [7,8]. Of particular interest is the lysine demethylase 3A (KDM3A; also known as JMJD1A and JHDM2A), a histone H3 mono- and di-methyl lysine-9 (H3-K9me1/2) Jumonji C (JmjC)-type KDM. KDM3A is upregulated in several cancers and coordinates with multiple oncogenic transcription factors (e.g., c-Myc, androgen receptor, estrogen receptor, β-catenin, hypoxia-inducible factor-1α, etc.) to promote cancer progression and therapeutic resistance [9]. In this manner, KDM3A upregulates transcription factor activity by removing the repressive H3-K9me1/2 marks. Intriguingly, the mechanisms behind the involvement of KDM3A in cancerous processes are now known to include the demethylation of non-histone proteins. Specifically, KDM3A was reported to demethylate p53-K372me1 and PGC-1α-K224me1 [10,11].

The number of non-histone or non-canonical KDM substrates has slowly been increasing. LSD1, an FAD-dependent KDM of the amine oxidase family, is known to demethylate more than 10 non-histone methylation sites thus far [12,13]. Besides KDM3A, several non-histone substrates for other JmjC KDMs have been reported, including JMJD1C, KDM2A, and the KDM4 family [14,15,16,17,18]. In comparison, the number of non-histone substrates of KMTs has dramatically increased during the past two decades. This is, in part, due to systematic explorations of enzyme substrate specificity, using peptide permutation libraries based on the canonical histone substrate. Specifically, the substrate specificity profile of several KMTs have been studied and include Dim-5, G9a, SET7/9, NSD1, and SMYD2 [19,20,21,22,23]. In the case of SET7/9, this approach led to the identification of a total of 91 new peptide substrates [21]. More recently, this approach was applied to a JmjC KDM, KDM5A, for the first time and successfully identified a number of new in vitro substrates [24].

In this study, we systematically explored KDM3A substrate specificity through the peptide permutation library and used this information to predict and validate novel substrates for further explorations. The peptide permutation library was based on the canonical KDM3A substrate, H3-K9me2. In this manner, each residue +/−4 positioned directly adjacent to the di-methylation site was individually exchanged to the other 19 naturally occurring amino acids (Figure 1A). After screening KDM3A activity against the permutation library, recognition motifs were generated, and peptides were scored at multiple levels of stringency (Figure 1B). Specifically, scoring and predictions are provided for peptides comprising the KDM3A interactome to identify potential substrates that could facilitate the role of KDM3A in cancer. Among these predictions, we successfully identified MLL1, p300, and KDM6B as in vitro KDM3A substrates. These substrates may be further explored in tissue culture experiments to decipher the avenues by which KDM3A could function as an oncogene.

## 2. Materials and Methods

### 2.1. Peptide Synthesis

Synthetic H3_2-16_ peptides were made at a scale of 2 µmol following standard Fmoc (n-(9-fluorenyl)methoxycarbonyl) chemistry on an automated ResPep SL peptide synthesizer (Intavis Bioanalytical Instruments, Köln, Germany). All Fmoc-protected amino acids, both standard and modified, were purchased from P3 BioSystems (P3 BioSystems, Louisville, KY, USA), with the exception of Fmoc-Lys(Me)3-OH Chloride (Sigma-Aldrich, St. Louis, MO, USA).

Peptides were synthesized essentially as previously described [26,27]. Briefly, peptides were synthesized from the C-terminus to n-terminus by sequential addition of amino acids through repetitive cycles. Each cycle consisted of; (1) Fmoc deprotection (20% piperidine in DMF), (2) addition of c-terminally activated Fmoc-protected amino acid (i.e., activated with coupling reagents; HBTU in the presence of N-methylmorpholine in DMF), and (3) blocking of unreacted amines (2% acetic anhydride in DMF). Once complete, the peptides were cleaved from the resin and of the protecting groups with an acidic cleavage solution (95% trifluoroacetic acid, 3% tri-isopropylsilane, 2% water). Peptides were precipitated and washed with cold (i.e., −20 °C) ether, resuspended in 1X PBS (137 mM NaCl, 2.7 mM KCl, 10 mM Na_2_HPO_4_·2H_2_O, 1.8 mM KH_2_PO_4_ containing 5% acetic acid (Anachemia, Lachine, QC, Canada), and pH was adjusted to 7 with a 2M NaOH (BioShop, Burlington, ON, Canada) solution, and stored at −20 °C.

Predicted KDM3A substrate peptides were synthesized with a C-terminal tryptophan, separated from the main peptide sequence by a 6-aminohexanoic acid (6-ahx) flexible linker. These peptides were quantified by tryptophan fluorescence using the low-volume 2,2,2-trichloroethnaol (TCE) assay [28]. Briefly, to 10 µL of either tryptophan amino acid standard (0.05–1 mM) or diluted peptide (20-fold diluted in PBS), 10 µL of 5% TCE (Sigma-Aldrich, St. Louis, MO, USA) solution (diluted in PBS, Sigma; Cat # T54801) was added. After 30 min of incubation under a 15 W UV-lamp, modified tryptophan fluorescence was read with a BioTek Cytation 5 microplate reader(BioTek Instruments, Winooski, VT, USA). To enable quantification based on tryptophan alone, and not tyrosine, fluorescence emission was measured at ʎ  =  515 nm with an excitation of ʎ  =  355 nm.

### 2.2. KDM3A Expression and Purification

*Spodoptera frugiperda* (Sf9; American Type Culture Collection) cells were cultivated in Graces Insect Media (Gibco) supplemented with 10% (*v*/*v*) fetal bovine serum (FBS; Gibco) and penicillin-streptomycin (Wisent Bioproducts) at 27 °C in a humidified chamber.

The baculovirus transfer vector, for downstream expression of recombinant KDM3A_515-1317_-His-Flag, was a gift from Nicola Burgess-Brown and was constructed as described elsewhere [29]. Briefly, this vector was transformed into DH10BAC-competent cells and the isolated bacmid was transfected into Sf9 cells to produce recombinant baculovirus (i.e., P1 virus). The virus was amplified to P2 then P3 generations by subsequent 10-day infections of Sf9 cultures at 1:100 virus-to-culture volume ratios. For large scale protein expression, Sf9 cells were seeded at 500,000 cells/mL in 250 mL of growth media in a spinner flask propelling at 130 rpm. Upon reaching 2,000,000 cells/mL, the suspension culture was supplemented with 250 mL of IMAX media (Wisent) and infected with a P3 virus at a 1:100 ratio. After 60 h, the cells were collected by centrifugation, washed with 1X PBS, and the cell pellet was snap frozen and stored at −80 °C.

Purification was performed via standard Ni-NTA chromatography as described elsewhere [29,30] with several modifications. All steps were performed at 4 °C. Briefly, the pellet was resuspended in lysis buffer (50 mM HEPES-KOH pH 7.4, 300 mM KCl, 5 mM imidazole, 5% (*v*/*v*) glycerol, 0.05% (*v*/*v*) Triton X-100), supplemented with Pierce Protease Inhibitor Tablets (Pierce). The resuspension was dounce homogenized (25 passes), sonicated at 40% amplitude for 3 cycles of 30 s on and 30 s off, dounce homogenization was repeated, and the lysate was clarified through centrifugation at 18,000 rpm for 45 min(Thermo Fisher Scientific, Waltham, MA, USA). The clarified lysate was incubated with Ni-NTA agarose beads (Qiagen) for 2.5 h with end-to-end rotation. Beads were washed with 100 bead volumes of wash buffer (lysis buffer containing 40 mM imidazole), and proteins were eluted multiple times with 1 bead volume of elution buffer (lysis buffer containing 250 mM imidazole). Fractions were dialyzed into storage buffer (20 mM HEPES-KOH, pH 7.4, 300 mM KCl, 5% (*v*/*v*) glycerol) using 10K MWCO Slide-A-Lyzer Dialysis Cassettes (Thermo Fisher Scientific, Waltham, MA, USA), flash frozen in liquid nitrogen, and stored at −80 °C.

### 2.3. SDS-PAGE and Coomassie Staining

Two microliters of purified KDM3A fraction was diluted with 2X Laemmli sample buffer (120 mM Tris-Cl, pH 6.8, 4% *w*/*v* SDS, 20% *v*/*v* glycerol, 5% *v*/*v* 2-mercaptoethanol), heated at 95 °C for 3 min, and cooled at room temperature. The sample and 3 μL Precision Plus Protein All Blue Standards (BioRad, cat no. 1610373) were subject to resolution on a standard SDS-PAGE gel (6% stacking and 8% resolving gel) at 150 V for 60 min. After resolution, the proteins were visualized following incubations in coomassie staining (1g/L Coomassie Brilliant Blue R250, 50% *v*/*v* methanol, 10% *v*/*v* glacial acetic acid, 40% *v*/*v* water) and destaining (50% *v*/*v* methanol, 10% *v*/*v* glacial acetic acid, 40% *v*/*v* water) solutions. The gel was imaged on a BioRadGel Doc XR+ Imaging System.

### 2.4. KDM3A Activity Assay

KDM3A activity towards peptides comprising the H3-K9me2 permutation library was monitored with the Succinate-Glo JmjC Demethylase/Hydroxylase Assay (Promega; Cat# V7990) with minor modifications. Briefly, 5 µL demethylase reactions (50 mM HEPES pH 7.0, 10 µM H3_2-16_ peptide, 10 µM Fe(II)SO_4_, 10 µM 2-oxoglutarate, 100 µM ascorbic acid, 0.5 mM TCEP, 10 µg/mL BSA, 1% *v*/*v* DMSO) were performed at 23 °C for 3 h. Subsequent detection of succinate was performed exactly as described by the manufacturer (Proemga, Madison, WI, USA). Briefly, 5 µL of Detection I was added to each demethylase reaction and incubated for 1 h, followed by the addition of 10 µL of Detection II and luminescence was read after 20 min with a BioTek Cytation 5 microplate reader. Screening of demethylase activity towards individual peptides within the permutated H3_2-16_-K9me2 library was performed with 600 nM of recombinant enzyme.

For screening of KDM3A activity towards peptides representing candidate substrate predictions, a modified procedure to the one described above was performed. Demethylase reactions were performed as described above except with the presence of 1% *v*/*v* Succinate-Glo Solution and 1% *v*/*v* Acetoacetyl-CoA (i.e., diluted 100-fold in the final reaction volume). In this manner, the succinate that was formed during the demethylase reaction was immediately converted. Following the demethylase reaction, 5 µL of Succinate-Glo Buffer was added to each demethylase reaction and incubated for 1 h. Luminescence was read after the addition of 10 µL of Detection II exactly as described above.

### 2.5. Peptide List

A list of KDM3A interactors was assembled by combining all human interactors displayed on NCBI—Gene and BioGRID [31]. Additionally, all PubMed articles containing “KDM3A”, JHDM2A”, “JMJD1A”, and “TSGA” were assessed for evidence of interactors not listed within these databases (i.e., data providing evidence for protein–protein interactions, existence within the same complex, interactions with modifying enzymes, etc.) (PubMed and databases accessed on 7 December 2020). Lysine-centered 9-mer windows representing all lysine residues within these interactors were extracted from protein sequences provided in the UniProtKB database [32]. All interactors and sequences are provided in Appendix A, and derived peptides are provided in Appendix A. Several lysine residues residing near n- or c-terminal ends, for which it was not possible to provide lysine-centered 9-mer peptides, were excluded from analysis (noted in Appendix A).

## 3. Results

JmjC KDM-catalyzed demethylation of the methyllysyl residues may be detected via several different techniques due to the production of demethylated peptide, the conversion multiple cofactors, as well as the formation of formaldehyde as a byproduct. Here, we monitored JmjC KDM activity by detecting the production of succinate via a luminescent-based assay [33]. Preceding demethylation, JmjC KDMs, like all the Fe(II)/2-oxoglutarate-dependent family of dioxygenases, produce succinate through the decarboxylation of 2-oxoglutarate. This conversion occurs more readily in the presence of substrate peptide; however, some uncoupled 2-oxoglutarate conversion may occur in the absence of peptide. To accurately assess relative levels of substrate demethylation, succinate formation in the absence of peptide should be used to establish an assay baseline.

### 3.1. KDM3A Methyl-State Preference

Prior to assessing the specificity of KDM3A for peptide sequences flanking the demethylation site, the methyl-state preference of KDM3A was first reconfirmed to determine the optimal H3-K9 methyl peptide for downstream permutation. Firstly, recombinant KDM3A purity and non-saturating enzyme concentration, for the demethylation activity assay, were confirmed (Figure 2A,B). Under non-saturating assay conditions, KDM3A demonstrated the highest level of demethylation towards H3-K9me2 peptide (Figure 2C). As expected, demethylation of H3-K9me1 was the second highest and H3-K9me3 was the lowest among the three methylation states. Furthermore, compared to the null methyl H3-K9 peptide, succinate formation in the presence of the H3-K9me2 peptide was the most significant (Figure 2C). Thus, the permutation library, synthesized for downstream steps, was based on the di-methylated version of the H3-K9 peptide.

### 3.2. KDM3A Substrate Specificity

To assess the substrate specificity (i.e., target sequence preference, recognition motif) of KDM3A, a peptide permutation library of the H3-K9me2 peptide was synthesized for in-solution assays. Each residue flanking (+/−4 residues) the di-methylation site was individually exchanged to the other 19 naturally occurring amino acids. In this manner, the relative effect of all 20 amino acids in each residue position site, flanking the di-methylation, could be assessed.

KDM3A demethylation activity monitored in the presence of each permutated peptide was determined relative to the wild-type H3-K9me2 peptide. The resultant substrate specificity of KDM3A was represented in both heat-map and motif formats for complete clarity (Figure 3). Without previous structural information on the interaction between KDM3A and H3-K9me2 peptide, it is difficult to discern expected trends. However, similar to the explorations of substrate specificity of other enzymes, specifically methyl-regulators, certain residue positions demonstrate high flexibility, whereas others demonstrate strict requirements. The strictest requirements for KDM3A demethylation of H3-K9me2 were the presence of an alanine residue at P-2 and a glycine residue at P-4. This is the case given that no amino acid substitution at these positions conferred at least 0.5 relative activity (i.e., 50% activity relative to the wild-type peptide) (Figure 3B). Notably, the presence of an alanine at the P-2 position was also shown to be a strict requirement for other methyl regulators, such as CBX1 and G9a, assessed by permutation and oriented-peptide library, respectively [34,35].

### 3.3. Substrate Prediction, Dataset Description, and In Vitro Validation

Studies assessing the substrate specificity of modifying enzymes, specifically methyl regulators, have primarily leveraged this information for substrate prediction. Traditionally, this has been performed through facile searches for the presence of recognition motifs within protein sequences using online tools, such as Scansite and ScanProsite [36,37]. However, only the canonical H3-K9 peptide sequence returns as an exact match using the three KDM3A recognition motifs (Figure 3B; low-, medium-, and high-stringency motifs). This is likely due to the high degree of strictness at several residue positions (P − 2, P + 2, P + 3, and P + 4). Thus, we sought an approach that may be more permissive to substrate discovery efforts. Rather than binary decision making (i.e., yes/no exact match), we stratified queried peptides for the degree to which each sequence resembled the recognition motif. In this manner, each peptide was given a ‘peptide score’, where the score reflects the number of residues in the peptide sequence matching those observed in the motif at each position (Figure 1B). Similarly, this has been applied to substrate predictions for KDM5A [24]. Specifically, a peptide list representing the KDM3A interactome was scored.

To enable scoring at multiple levels of stringency, multiple KDM3A recognition motifs, defined by different activity thresholds, were used. Specifically, the 0.5, 0.75, and 1.0 threshold motifs were used for low-, medium-, and high-stringency scoring (Figure 3B). For scoring purposes, these recognition motifs were represented as position-specific scoring matrices (PSSMs; Appendix A). Each PSSM was used to score the lysine-centered 9-residue windows, representing the full KDM3A interactome (4880 peptides). Due to the size of this dataset, only the top peptides (i.e., ≥3 standard deviations above the mean) (Figure 4A) from scoring with a low-stringency motif (i.e., 0.5 threshold) are provided in Appendix A. Scoring of the complete KDM3A interactome with each recognition motif (low-, medium-, and high-stringency) is provided in Appendix A.

In aiming to validate the substrate predictions, we monitored KDM3A activity towards 15-mer peptide substrates, representing all top-ranked windows that were scored with the low-stringency motif (Appendix A). Given that the motif used for scoring represents amino acid substitutions permitting at least 50% wild-type H3-K9me2 activity, we applied the same threshold to define validated substrates. Among the 16 predictions, 3 peptides were found to meet this criterion (Figure 4B). Thus, MLL1-K1534me2, KDM6B-K991me2, and p300-K1774 were identified as in vitro KDM3A substrates. Furthermore, we sought to explore whether these peptides share any sequence features with the H3-K9me2 peptide outside of the +/−4 residue window originally assessed. Individually, the three substrates share some features (Appendix A). For example, MLL1-K1534 and KDM6B-K991 peptides both have a proline residue at the P + 7 position, just as the H3-K9 peptide. Notably, p300-K1774, MLL1-K1534, and H3-K9 peptides share a ‘TK’ sequence at the P-6/P-5 positions.

## 4. Discussion

Substrate specificity analysis using systematically designed peptide libraries has been a valuable tool for exploring protein–protein interactions, specifically for substrate discovery efforts [38]. In the field of lysine methylation, methods involving the permutation of canonical peptide substrates have primarily led to the discovery of non-canonical substrates for KMTs and was recently applied to a JmjC KDM for the first time [24]. Given the significance of KDM3A in cancer, here, we applied this approach to study the substrate specificity of this JmjC KDM and thereby aid in future substrate discovery explorations.

The substrate specificity of KDM3A, as determined by permutation of the H3-K9me2 peptide, shows that several residue positions are highly flexible, whereas others are strict regarding tolerable amino acid substitutions (Figure 3). Without prior structural information on contacts between this enzyme-peptide interaction, in-depth mechanistic hypotheses explaining the effect of residue substitutions cannot be made. However, the presence of strict sequence requirements suggests that the enzyme performs with specificity in this in vitro environment, which is in line with permutation-based explorations of other methyl regulators. Additionally, KDM3A activity is known to not be promiscuous in nature, given that early studies on this enzyme demonstrated demethylation activity only towards H3-K9 and no other histone H3 and H4 sites that were tested [14]. Finally, KDM3A shares substrate specificity features with other H3-K9 methyl regulators. The methyl-binding domain CBX1 and the G9a KMT have both been shown to rely on the presence of an alanine residue at the P-2 position [34,35]. One complexity to consider is that the template sequence used for the peptide permutation library may influence the observed specificity profile. For example, it was reported that the JMJD2A-double Tudor domains display distinct specificities for peptide sequence between three different permutation arrays (H3-K23me3, H3-K4me3, and H4-K20me3) [34]. Whether this is the case for KDM3A or other methyl regulators requires further exploration. Altogether, KDM3A exhibits clear preferences for peptide sequences that are requisite for demethylation of the H3-K9me2 peptide.

In aiming to provide a set of high-confidence substrate predictions, KDM3A substrate recognition motifs were used to stratify queried peptides for the degree to which these sequences resemble the recognition motif. In this manner, ‘peptide scores’ represent the number of residue positions in the queried peptide with amino acids matching those in the recognition motif. We decided to explore all lysine residues on proteins with a known functional relationship with KDM3A (i.e., the KDM3A interactome). Complete scoring is provided for this peptide list at each level of stringency (Appendix A).

To be more permissive towards substrate discovery efforts, a set of high-confidence predictions were identified using a low-stringency KDM3A recognition motif (Appendix A). With peptide scores higher than 3 standard deviations above the population mean, the queried peptide sequences match with at least seven of the nine residue positions in the recognition motif. In screening for KDM3A activity towards these 16 candidates, we successfully identified 3 new in vitro peptide substrates of KDM3A; MLL1-K1534me2, KDM6B-K991me2, and p300-K1774me2.

Similar to KDM3A, the newly identified substrates are all chromatin-associated proteins, given that they are epigenetic enzymes. KDM6B (also known as JMJD3) is an H3-K27me2/3 demethylase, which was first found to activate expression of the *HOX* gene (a developmental-related gene) through its activity [39]. The K991 position resides in an intrinsically disordered region of the protein, though, at this time, the function of this residue is not known. However, the KDM6 subfamily is known to have aberrant functions in human cancers [40]. Research on the role of KDM6B in cancer has primarily demonstrated the role of this enzyme as a tumor promotor, through the function of this enzyme is context-dependent, as this enzyme mediates bother carcinogenic and anti-cancer signaling pathways (reviewed by Hua and colleagues [40]). Relevantly, radiotherapy resistance of esophageal squamous cell carcinoma, promoted by hypoxia, has been shown to occur through increased expression of KDM3A and KDM6B [41].

Notably, although the p300-K1774 site was identified as a high-ranking candidate using the low-stringency recognition motif, we also found that this window scored the highest (other than H3-K9me2) when using the high-stringency recognition motif (matching with seven out of nine residue positions in the motif; Appendix A). p300 (also known as EP300) is a lysine acetyltransferase known to modify both histone and non-histone proteins, whereby the former event is well recognized in promoting target gene expression [42]. The K1774 residue is known to be acetylated and this modification was first identified in a study assessing autoacetylation of p300 [43], though the functional significance of K1774 acetylation is unknown. However, this site resides in the C-terminal transcriptional adaptor zinc-binding domain (i.e., Taz2), and, thus, may be involved in the interaction of p300 with numerous transcription factors and other regulators. Interactors of the p300 Taz2 domain include, but are not limited to, the E1A oncoprotein [44], p53 [45,46], B-Myb [47], MEF2 [48], and STAT1 [49]. It is interesting to note that KDM3A and p300 have been shown to physically associate in colorectal cancer [50]. Specifically, KDM3A was found to be a required factor for p300 recruitment to the enhancers of hippo target genes. It would be interesting to see whether KDM3A-mediated demethylation of p300-K1774 is involved in these findings or regulates p300 Taz2-mediated protein–protein interactions.

Finally, Mixed lineage leukemia protein-1 (i.e., MLL1, also known as KMT2A) is an H3-K4 methyltransferase, well known for its involvement in a plethora of chromosomal translocations occurring in leukemia [51,52]. The function of the MLL-K1534 residue is currently unknown; however, this residue resides directly next to the plant homeodomain 2 (PHD2) finger. The PHD fingers of MLL1-5 proteins have a diverse array of biological functions (reviewed by Ali and colleagues [53]). The PHD2 finger of MLL1 has been shown to possess E3 ubiquitin ligase activity and this domain negatively regulates MLL1 transcriptional activity [54]. Specifically, MLL1-PHD2 mutants were more stable than WT MLL1 and showed increased recruitment to the promotor region of a Hoxa9 reporter. Furthermore, the MLL1-PHD2 finger was found to be necessary for homodimerization of the PHD1-3 region [55]. It would be interesting to test whether the K1534 site is involved in any of these activities. Notably, the role of KDM3A in colorectal cancer also directly involves MLL1. Specifically, in studying the role of KDM3A/B in human colorectal cancer stem cells (CSCs), it was found that these demethylases recruit MLL1 to specific gene promotors to facilitate Wnt target gene activation [55]. Furthermore, the same study showed that MLL1 physically interacts with KDM3A and KDM3B.

In aiming to extract additional information regarding the newly identified KDM3A substrates, we asked whether the 15-mer peptides used for KDM3A activity assays may be present in other proteins. The full 15-mer MLL1-K1534 peptide was found in two MLL1 fusion proteins (CDK6/MLL, GenBank: AAM33377.1; KMT2A-ELL, GenBank: QSQ01632.1). Furthermore, the p300-K1774 peptide is similar to a homologous region on CREB-binding protein (CBP; Appendix A). The peptide is identical in every residue position, except the final C-terminal residue has a valine in CBP instead of an isoleucine in p300. Given the biochemical conservation between these residues, it would be interesting to see if KDM3A may also demethylate CBP-K1811, a residue also occurring in the Taz2 domain of CBP.

Lastly, to examine whether these newly identified substrates are functionally tied together, we performed STRING analysis to search for known protein–protein associations [56]. Given that KDM6B, p300, and MLL1 all associate with chromatin proteins, it was unsurprising to find that these epigenetic regulators are interconnected among themselves and share multiple common interactors (Figure 5; for simplicity, only five common interactors are shown). Furthermore, KDM6B is known to interact with common MLL complex proteins [57]. Thus, through the logic of “guilt by association”, it is plausible that the role of KDM3A in cancer may extend to involving interactors of the newly identified substrates in this study.

As mentioned, the screen of KDM3A activity with the H3-K9me2 peptide permutation library revealed that this enzyme behaves with some level of specificity in an in vitro environment. However, it should be appreciated that peptides that were not permitting KDM3A activity may be missing essential elements that would otherwise be present in a cellular context. Furthermore, it should also be recognized that further characterization, in cells, on protein substrates is required to validate the new peptide substrates shown here and evaluate the biological relevance. This validation would rule out false-positive leads that may only appear ‘true’ in an in vitro setting, involving short peptide substrates. Several approaches may be taken to validate these findings, besides the generation and use of custom methyl- and site-specific antibodies. Similar to the validation of H2B-K43me2 as a KDM5B demethylation site, substrate proteins may be purified from mammalian cells and then subject to in vitro demethylation, or purified after manipulating cellular KDM3A levels, and subject to targeted-mass spectrometry [58]. Though, the cellular conditions must be specific, such that they permit these proteins to carry these specific methylation marks. Nonetheless, these new in vitro substrates provide testable hypotheses for the oncogenic role of KDM3A.

## 5. Conclusions

To conclude, this study (1) uncovered the substrate specificity of KDM3A using an H3-K9me2 peptide permutation library and (2) identified MLL1-K1534me2, KDM6B-K991me2, and p300-K1774me2 peptides as in vitro substrates of KDM3A. The former may be used to deepen the understanding of the KDM3A interaction with H3-K9. The latter provides preliminary data, which may be further explored in tissue culture experiments, to decipher the mechanisms behind the role of KDM3A in cancer.

## Figures and Tables

**Figure 1 biomolecules-12-00641-f001:**
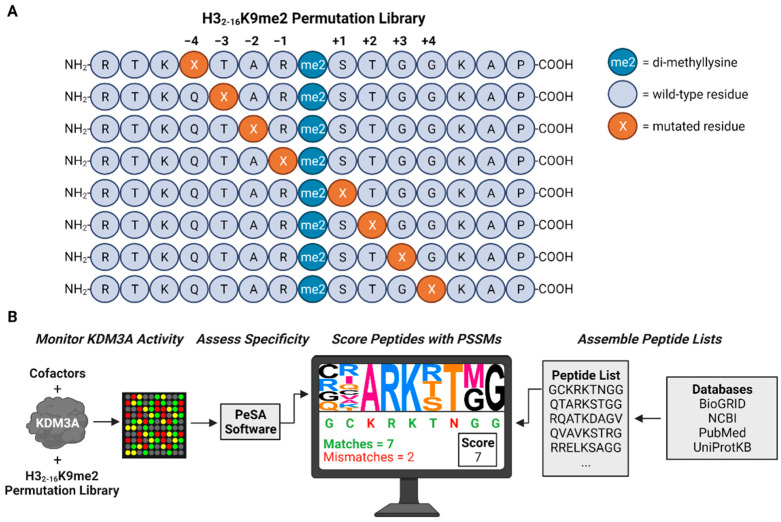
Overview of peptide permutation library and substrate prediction pipeline. (**A**) Permutated H3_2-16_ peptides di-methylated at the lysine-9 residue were designed such that each residue positioned +/−4 residues relative to the fixed di-methylation site was individually substituted to all other naturally occurring amino acids while the remainder of the sequence was unaltered. (**B**) Demethylase activity was monitored towards the H3_2-16_-K9me2 peptide permutation library, PeSA was used to visualize motifs and generate the corresponding position-specific scoring matrices (PSSMs) that were used to score peptides [25]. Peptides lists encompassing the methylproteome and KDM3A interactome were derived from several publicly available databases. The peptide score reflects the number of residues in the queried sequence matching those within the motif at each corresponding position. This figure was made in ©BioRender (biorender.com (accessed on 13 March 2022)).

**Figure 2 biomolecules-12-00641-f002:**
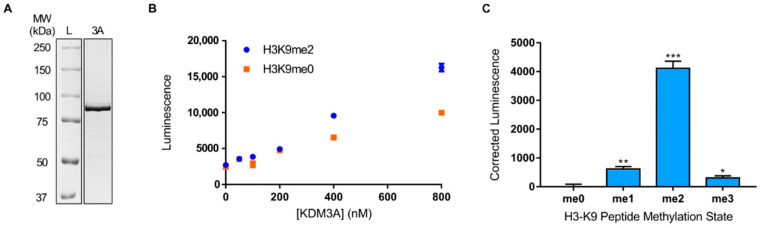
Confirmation of KDM3A purity, non-saturating enzyme concentration, and methyl-state preference. (**A**) Purity of recombinant KDM3A shown via Coomassie staining. (**B**) Titration of recombinant KDM3A enzyme using 10 µM of H3_2-16_-K9 null (orange) and di-methylated (blue), 10 µM Fe(II)SO_4_, 100 µM ascorbic acid, and 10 µM 2-oxoglutarate at 23 °C for 3 h. (**C**) Validation of KDM3A methyl-state preference. KDM3A methylation state preference using 10 µM of H3_2-16_K9 peptides of varying methylation states in the presence of 600 nM enzyme, 10 µM Fe(II)SO_4_, 100 µM ascorbic acid, and 10 µM 2-oxoglutarate at 23 °C for 3 h. Asterix denotes significance compared to me0 peptide where * *p* ≤ 0.05, ** *p* ≤ 0.01, and *** *p* ≤ 0.001.

**Figure 3 biomolecules-12-00641-f003:**
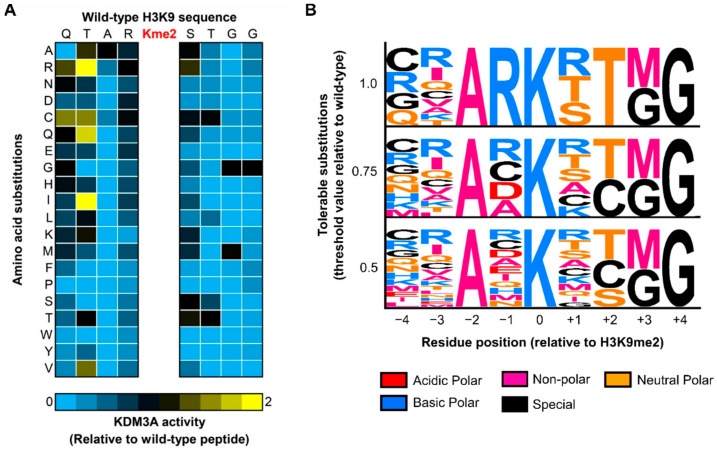
Substrate specificity profile of KDM3A towards H3_2-16_-K9me2 permutation library. (**A**) Activity of 600 nM KDM3A was assessed towards 10 µM of H3_2-16_-K9me2 WT or mutated peptide in the presence of 10 µM Fe(II)SO_4_, 100 µM ascorbic acid, and 10 µM 2-oxoglutarate for 3 h at 23 °C. On the H3 WT sequence, substitutions to all other naturally occurring amino acids took place +/−4 residues adjacent to the K9 di-methylation site. The WT H3 sequence and amino acid substitutions are shown on the horizontal and vertical axes, respectively. (**B**) PeSA-generated motifs representing the KDM3A substrate recognition motif at various thresholds (i.e., activity observed for a given peptide relative to the wild-type peptide). Motifs depict amino acid substitutions retaining relative activity greater than or equal to the defined threshold.

**Figure 4 biomolecules-12-00641-f004:**
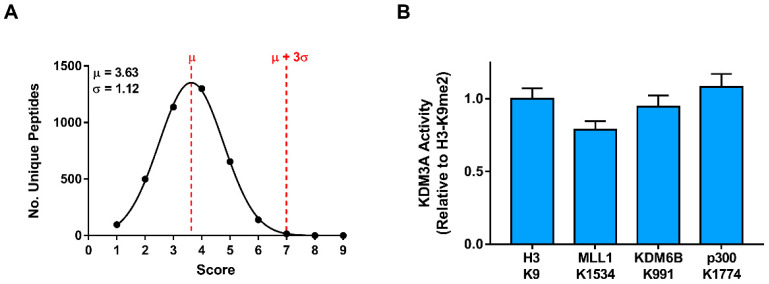
Substrate prediction and validation. (**A**) Distribution of scored peptides within the KDM3A interactome. Distributions represent scoring with a low-stringency KDM3A recognition motif (i.e., 0.5 threshold). All peptides ranking greater than or equal to three standard deviations (σ) above the mean (µ) score are provided in Appendix A. Gaussian curves were fitted with GraphPad Prism. (**B**) Identified KDM3A in vitro substrates displaying at least 50% relative H3-K9me2 demethylation. KDM3A activity was monitored towards K-centered 15-mer peptides, where the central di-methylated lysine residue is shown on the *x*-axis.

**Figure 5 biomolecules-12-00641-f005:**
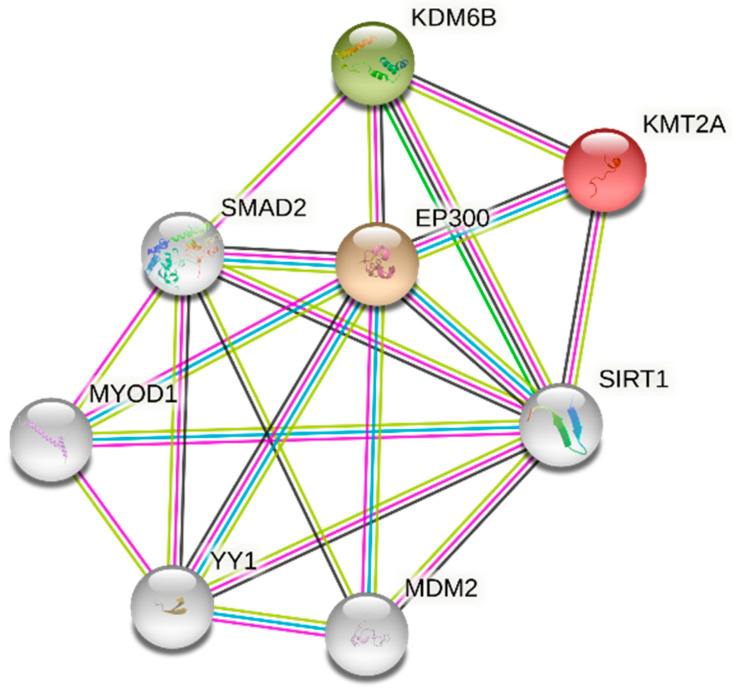
Protein–protein association network of newly identified KDM3A substrates. STRING-generated network showing the identified KDM3A substrates (colored nodes) and associated proteins (white nodes) [56]. Multiple types of interactions are shown: in magenta, experimentally determined known interactions; cyan, known interactions from curated databases; bright green, gene neighborhood; lime green, textmining; black, co-expression. A tabular form of this network is depicted in Appendix A.

## Data Availability

All relevant data are provided within the published manuscript and the Appendix A files.

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
