# Peer review of "Insights into a Cancer-Target Demethylase: Substrate Prediction through Systematic Specificity Analysis for KDM3A"

_biomolecules, 2022, doi:10.3390/biom12050641_

Round 1

Reviewer 1 Report

Attached files

Reviewer 2 Report

Chopra et. al., deciphered oncogenetic functions of KDM3A via identifying substrates starting with a peptide permutation library. They established KDM3A recognition motif and identified three substrates, which are all related to cancer.  The article is recommended for publication. 

Following minor changes are suggested. 

- It would be good to discuss any limitations or highlight any false positive leads experienced in the identification of substrates

Fig. 5A: Tabular form or any other graphical representation of the figure will add more clarity

Reviewer 3 Report

The study by Chopra et all uncovers 3 novel substrates for KDM3A, which were not described before with possible implications in cancer development. Overall, the paper is well written, organized and the conclusions are supported by the data presented. I only have a few minor suggestions to improve the manuscript:

  1. Please add in the material and methods section the method used for the results presented in Figure 2A.
  2.  The Supplementary Table 2 was not provided. Please add.
  3. Please add as future perspectives the next experiments to validate these findings. 
  4. Typos in page 5, line 166 - Following and page 7 line 259 decision.
